# ADVANCES IN SPARSE NEURAL MODEL TRAINING

## ABSTRACT

While larger neural models are pushing the boundaries on what deep learning can achieve, often more weights are needed to train models rather than to run inference for tasks. This paper explores how can we remove weights during training without impacting the ability to traverse parameter spaces (or the set of all weights a model can take). We first discover weights pruned for inference do not learn meaningful information over time but their short-term behavior is necessary for training. We then provide recommendations for removing such weights in order to train sparse neural models. With these recommendations, we attain competitive scores across dozens of deep learning workloads. We also find sparse models are tolerant of structures targeting existing hardware, opening avenues for training and inference acceleration. Our work encourages research to explore beyond massive neural models being used today.

## 1 INTRODUCTION

In the area of deep learning, increasing the size of neural models has led to dramatic advances (Hestness et al., 2017; Kaplan et al., 2020; Henighan et al., 2020), motivating training with hundreds of billions of parameters (Brown et al., 2020; Fedus et al., 2021). However, larger models incur higher memory costs and runtimes, which limits training and inference tasks that can run on existing hardware (Thompson et al., 2020). Nevertheless, larger models have been shown to train faster (Li et al., 2020) and better (Kaplan et al., 2020), which drives their adoption.

An area of research that seeks to compensate the increasing costs of model size is sparsity (Hoefler et al., 2021), which moves weights to zero so they can be discarded from storage or computations. Sparsity emerges as a promising option to reduce costs of inference (Luo et al., 2017; Allen-Zhu et al., 2019), as neural models are capable of performing tasks on sizes smaller than they were trained for (Narang et al., 2017; Renda et al., 2020). However, since overparameterization remains critical for training, sparsity observes limited success there (Lee et al., 2019; Frankle & Carbin, 2019; Wang et al., 2020; Tanaka et al., 2020; Frankle et al., 2021; Bellec et al., 2018; Mocanu et al., 2018; Dettmers & Zettlemoyer, 2019; Evci et al., 2020a; Jayakumar et al., 2020).

The ability to run inference but not training with fewer weights points to a problem with search (or how well can models navigate parameter spaces during training) rather than model capacity (LeCun et al., 1990). Literature shows adding more weights to training creates extra degrees of freedom that form new paths for optimization, rendering neural model training more effective (Evci et al., 2020b). While it is widely believed that having more weights facilitates escape from critical points during training, the question remains whether these weights are always needed (e.g., learn meaningful representations) or could be eventually discarded if they only help with the optimization process.

In this paper we explore how to remove weights from training without restricting its ability to effectively traverse parameter spaces towards good solutions. Surprisingly, we discover that weights pruned at inference time are transient and do not learn representations in the long-term. As such, these weights do not need to be stored throughout all of training, in contrast to weights that learn meaningful information for inference and thus need to be stored. The pruned weights however do learn short-term representations that provide the model extra degrees to escape regions of bad saddle points (Kawaguchi, 2016) and high error plateaus (Dauphin et al., 2014) that can slow down learning, after which they can be discarded.

With this understanding we propose new recommendations for training sparse models that consolidate recent advances in sparsity literature: (1) rewire weights to expand the parameter space, (2)

update gradients of all weights to encourage alternate paths of optimization, and (3) discard intermediate representations to reduce noise from gradient accumulations. Following these recommendations, we show that sparse neural models achieve competitive results on dozens of deep learning workloads, even when satisfying constraints needed to accelerate training and inference using Sparse Tensor Cores (Mishra et al., 2021) in NVIDIA GPUs.

The paper is organized as follows. In Section 2 we introduce a metric for learning and investigate the duration which weights retain meaningful information throughout training. We then devise recommendations for removing weights that do not learn long-term representations when training sparse models in Section 3. Section 4 summarizes our methodology for sparse training. Section 5 shows sparse models constructed in this fashion perform competitively across a plethora of deep learning workloads and can target current hardware accelerators (Mishra et al., 2021) for training and inference. Section 6 relates to prior research. In Section 7 we conclude with directions for future work.

## 2 METRIC FOR LEARNING

The crux of this paper is to understand whether weights that are being added for training are learning meaningful representations or solely being used to facilitate the optimization process. For this purpose, we use correlations to measure the duration that weights in a neural model retain (or learn) information that is relevant for inference.

Correlations determine how similar are two sets of data. When applied to weights learned over time, correlations measure how weights in future training time steps are similar to past ones. We postulate correlations represent the degree with which weights in a neural model learn. Weights that learn meaningful representations for inference are repeatedly reinforced throughout training, and their values form distinct temporal patterns that are correlated to past values. In contrast, weights that do not learn (and can be removed during inference) exhibit random behavior and have no dependence over time.

We measure correlations between a pair of time series $x$ and $y$ using the Pearson coefficient (Rodgers & Nicewander, 1988): $\rho_{x,y} \equiv E[(x - \mu_x)(y - \mu_y)]/\sigma_x \sigma_y$, where $\mu$ is the mean, $\sigma$ the standard deviation, and $E[\cdot]$ the expected value. Coefficients of value $+1$ denote two series have identical trends, 0 indicates the series are random, and $-1$ represents series with opposite behavior. We treat each individual weight as a time series $w \in \{w_1, w_2, \ldots, w_t\}$ and compute Pearson coefficients between windows $x \in \{w_1, w_2, \ldots, w_{t-\tau}\}$ and $y \in \{w_\tau, w_{\tau+1}, \ldots, w_t\}$, representing the similarity between the weights and their values at some future time $\tau$. Correlations naturally decay with $\tau$ since temporal patterns are less likely to persist over long durations. An important metric we adopt is the time $\Delta$ after which weights no longer correlate to their future values, namely when $\rho = 0$.

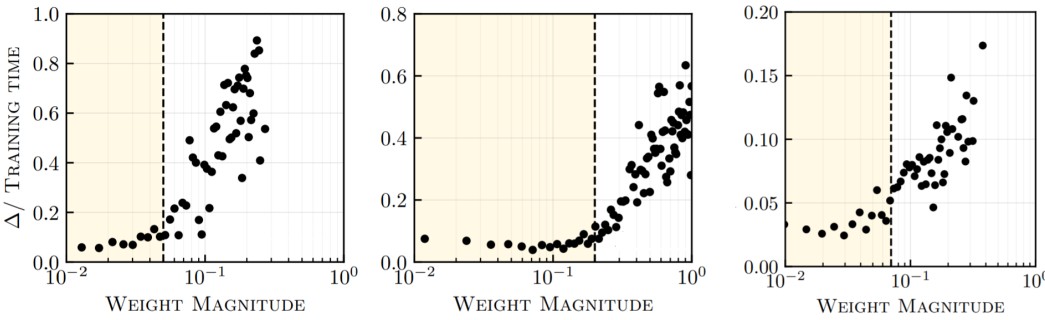

Figure 1: Time ($\Delta$) it takes for weights to decorrelate (normalized by the total number of training steps) as a function of weight magnitudes obtained after training. Points correspond to the median $\Delta$ over weight bins sampled from a single neural layer. Shaded regions distinguish between weights that are needed (white) or can be removed (yellow) at inference time. From left to right: Transformer-XL (Language Modeling), GNMT (Machine Translation), ResNet50 (Image Classification).

Figure 1 plots the time ($\Delta$) after which weights become completely uncorrelated with their past values. We observe that the correlation length grows with weight magnitudes obtained after training. While weights needed for inference exhibit correlations that persist across a significant fraction of training, weights that are not needed have short-term correlations and behave randomly over short periods. As a result, long-term correlations of large weights are characteristic of learning, signifying repeated reinforcement along the weight direction. On the other hand, the near-random motion of small weights suggests they don't learn useful representations and can be removed during inference.

Because weights removable during inference are absent of correlations over long periods of training, we conjecture they do not need to be stored all the time. However, we still need pruned weights during training due to their short-term correlations. More specifically, their short-term interactions are critical for training to take different paths for optimization, facilitating escape from bad saddle points (Kawaguchi, 2016) or high error plateaus (Dauphin et al., 2014) that can slow down learning. Combining this understanding, in later sections we discover their learned values can be periodically destroyed throughout training (once correlations fall to zero) with no impact on accuracy, which paves the road for sparse training.

## 3 RECOMMENDATIONS FOR SPARSE TRAINING

This section outlines recommendations for training sparse neural models. While many of these ideas have been circling in sparsity literature, they remain misunderstood which hinders adoption. We use the understanding from the previous section to provide a deeper intuition in these cases. To summarize, we uncover the following steps to train sparse neural models:

(1) Frequently rewire weights that participate in training.

(2) Perform gradient updates for weights that do not participate in training.

(3) Induce exploitation (e.g., stop rewiring after some amount of time, reset non-participating weights to zero, or regularize them).

### 3.1 REWIRING OF NEURAL WEIGHTS

Sparse models are commonly trained by rewiring (or sampling) a subset of weights from a neural model according to some criteria (magnitude, sign, gradients, etc), allowing them to learn which weights are needed for inference. Some works rewire weights every training iteration (Bellec et al., 2018; Wortsman et al., 2019; Raihan & Aamodt, 2020; Jayakumar et al., 2020; Zhou et al., 2021), while others rewire every hundreds of training steps (Evci et al., 2020a) or after an entire pass through the data (Mocanu et al., 2018; Dettmers & Zettlemoyer, 2019). Since there are no clear reasons behind these choices, an important question becomes how often must weights be rewired so neural models can learn effectively.

We study the effects rewiring rates have on accuracy when training sparse models of various sizes $d$, where $d$ denotes the fraction of total weights being used in the sparse model. Figure 2 (left) plots the task error (or accuracy difference between dense and sparse models) as a function of the number of training steps $r$ taken between rewirings. We find rewiring frequency does not matter when sparsity is low or moderate ($d \sim 0.5$), but errors increase with $r$ as models get sparser ($d \ll 0.5$). Therefore, weights should be rewired frequently (within a few training steps) to avoid losing accuracy.

Correlations can help us explain why training improves when we rewire sparse neural models. While weights that do not learn long-term representations can be discarded during training, they have short-term correlations that create extra degrees of freedom for escaping critical points (Evci et al., 2020b). But this escape only occurs when weights are rewired. For example, neural models always operate on reduced parameter spaces if weights being used remain unchanged throughout training ($r \to \infty$), and expand exactly once after going through the entire data when they are rewired every epoch. By swapping between weights that participate (are used in forward and backward propagations) and do not participate (are fixed to zero during training), training can take different paths for optimization using the short-term correlations, which helps neural models explore the parameter space.

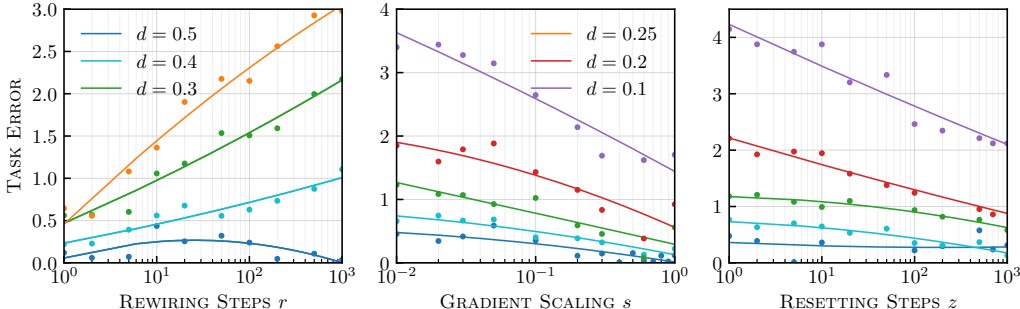

Figure 2: Investigations into various aspects of training sparse models using ResNet50. Left: Task error (or accuracy difference between dense and sparse models) as a function of rewiring steps $r$. Middle: Task error as a function of scaling factor $s$ applied to gradients of weights that do not participate in training. Right: Task error where non-participating weights are reset to zero every $z$ training steps. We consider sparse models of different $d$ denoting the ratio of weights being used. Lines represent polynomial fits of sample points. Appendix A covers data for a broader span of neural models and tasks.

## 3.2 UPDATES FOR WEIGHTS NOT PARTICIPATING IN TRAINING

Previously, we found neural weights not needed for inference form short-term interactions that make training more effective. Since sparse models lack weights that have similar roles, we can approximate this behavior using gradient updates for non-participating weights. Because non-participating weights do not contribute to the loss, their gradients determine whether another optimization path is better suited to traverse the region being explored in the loss landscape. Repeating gradient updates registers the importance of these paths over some period of training and can trigger rewiring when weights exceed a threshold, which allows training to explore different regions of the parameter space while operating on (or forward and backward propagating) a smaller set of weights.

We evaluate the importance of gradients updates on non-participating weights by reducing their contribution with a scale factor $s$. Figure 2 (middle) shows the task error as a function of the scale factor $s$ using various model sizes $d$. We observe error increases as $s \rightarrow 0$ (no gradients contribute when $s = 0$), which can be attributed to premature convergence when training lacks expressive power to explore different paths. Dampening the gradient updates affect sparser models ($d \ll 0.5$), which are more sensitive to critical points, more than larger models ($d \sim 0.5$), which may still have sufficient weights to train effectively. As a result, non-participating weights should be updated regularly to retain accuracy.

Gradient updates for weights that do not participate in training could mean they also learn representations, such that more capacity rather than search improves accuracy. We verify this by resetting non-participating weights to zero after every $z$ training steps, thus removing any representations they might have learned. Figure 2 (right) shows the task error as a function of $z$. Errors are the highest when values are reset every iteration ($z = 1$), which is equivalent when training without any rewiring. Conversely, error decreases as weights are reset less frequently and saturates after sufficient training steps ($z \sim 1k$). Interestingly, this also represents the time it takes for added weights to decorrelate, as shown in previous sections, supporting our conjecture that correlations are indicative of learning, and weights can be discarded once they cease to exist. The ability to remove information from non-participating weights reinforces the idea that they make training more effective rather than augment model capacity.

While most of literature trains sparse models using fewer weights and gradients (Bellec et al., 2018; Mocanu et al., 2018; Dettmers & Zettlemoyer, 2019), more recent success has been found by updates gradients for weights that do not participate in training (Wortsman et al., 2019; Liu et al., 2020; Zhou et al., 2021; Hubara et al., 2021) as described above. However, so far there has been little understanding of why this helps improve sparse training, which we attribute to the use of short-term correlations to escape critical points.

### 3.3 BALANCING BETWEEN EXPLORATION AND EXPLOITATION

Deep learning like other optimization problems treads a delicate balance between exploration and exploitation. In early stages of training neural models explore search spaces using high learning rates, whereas late stages exploit specific regions using small learning rates. Training sparse neural models can also affect this balance, as fewer weights reduces the degrees of freedom and thus hinders exploration during training. On the other hand, gradient updates on non-participating weights introduces noise as any nonzero value will not reflect what is being used for training, which limits exploitation (training bounces around basins of attraction due to gradient noise).

Figure 3 shows task error degrades without proper exploration, by training smaller models with neural layers of reduced widths (NO EXPLORE), or exploitation, by training sparse models delineated in previous sections (NO EXPLOIT). Therefore, we seek ways to induce exploitation when augmenting search spaces for more exploration. One course of action is to remove noise introduced by non-participating weights during late stages, so training can take steepest descents towards the minima. To achieve this we stop rewiring weights after sufficient training (FIX), such that non-participating weights can no longer contribute to the loss. Figure 3 shows this decreases error rates tremendously.

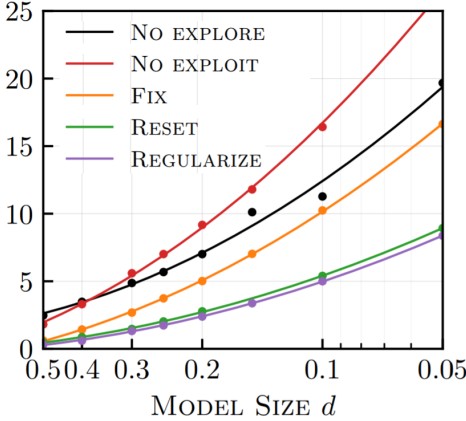

Figure 3: Task error after training Transformer-XL Base using different exploration versus exploitation strategies as a function of model size $d$. Appendix A covers data for a broader span of neural models and tasks.

Another option is to draw analogies with added weights, which can be safely discarded after sufficient training since they do not learn representations over time. We reset non-participating weights to zero roughly every $1k$ training steps (or the time it takes for added weights to decorrelate) in order to remove gradient noise that may trigger unnecessary rewiring, allowing training to exploit nearby regions. Figure 3 shows error decreases substantially when non-participating weights are either reset to zero (RESET) or regularized with a decay factor (REGULARIZE) as suggested in (Zhou et al., 2021). Sparsity literature introduces alternative courses for inducing exploitation (Mocanu et al., 2018; Dettmers & Zettlemoyer, 2019; Evci et al., 2020a).

## 4 METHODOLOGY

Using recommendations from the previous section, we summarize a methodology for sparse training in Algorithm 1. Namely, we rewire weights $w$ of sparse neural models based on their magnitudes in order to preserve long-term correlations that were discovered in Section 2. At each point in time, the top fraction $d$ of weights are chosen to participate in training. Participating weights $p$ are used to compute the loss and activation gradients, while the optimizer performs updates on all the weight gradients. We then induce exploitation on non-participating weights $n$ using any of the approaches discussed earlier.

We note some limitations of the methodology described above. Sparse training observes no memory savings because all weights need to be stored for their gradient updates. Sparsity is also only applied on weights, which means we can only accelerate the forward pass and part of the backward pass that computes activation gradients. For further acceleration, we can consider applying sparsity to activations or their gradients to accelerate weight gradient computations (Raihan & Aamodt, 2020).

The methods delineated above make a tradeoff between exploration and exploitation using variables $v$, $z$, and $\beta$. Stopping rewiring after half of training ($v = 0.5$) often provides the best results. $z$ should be large enough to retain short-term correlations (on the order of $1k$ training steps). $\beta = 0.0002$ chosen in (Zhou et al., 2021) roughly matches our choice for $z$. All methods perform equally

well with their optimal variables, so these variables can either be automated or kept as default. For most experiments we adopt RESET and use FIX in a few select cases.

---

**Algorithm 1** TRAIN($\mathcal{D}, \gamma, d, v, z, \beta$)

---

1: Initialize neural weights $w$ at random
2: **for** each training iteration $t$ **do**
3:     Sample a mini batch of data $\mathcal{D}$
4:     $p \leftarrow w_i$ if $w_i \geq \tau$, where $\tau$ is chosen so that $|p| = d$     $\triangleright$ Get participating weights
5:     $n \leftarrow w_i$ if $w_i < \tau$            $\triangleright$ Get non-participating weights
6:     $\ell \leftarrow L(p, \mathcal{D})$              $\triangleright$ Forward pass
7:     $\frac{\partial \ell}{\partial w} \leftarrow \frac{\partial L(p, \mathcal{D})}{\partial w}$            $\triangleright$ Backward pass
8:     $w \leftarrow w + \gamma \frac{\partial \ell}{\partial w}$            $\triangleright$ Optimizer step
9:     **if** $t \geq v$ **then** $n \leftarrow 0$           $\triangleright$ FIX
10:    **if** $t \bmod z = 0$ **then** $n \leftarrow 0$        $\triangleright$ RESET
11:    $n \leftarrow n - \beta n$             $\triangleright$ REGULARIZE

---

## 5 EMPIRICAL DATA

We are now in the position to train sparse neural models using our recommendations. This section presents empirical evidence that short-term correlations can help sparse models achieve better accuracy for inference tasks. We first demonstrate that our strategy performs competitively against state-of-the-art sparsity research. Then, we explore sparse models are tolerant to sparsity structures that target hardware acceleration using Sparse Tensor Cores (Mishra et al., 2021).

### 5.1 COMPARISONS TO DENSE MODELS

Experiments are conducted across a wide range of deep learning tasks and neural architectures trained on large data sets, as detailed in Appendix B. We draw comparisons between dense models (DENSE) and sparse models as described in this work (SPARSE). We also compare to smaller dense models (SMALL) consisting of neural layers with reduced widths that can run on existing dense hardware. We reduce the widths of neural layers by a factor of $1/d$ along one of the dimensions to match the number of weights used in the sparse models. Appendix C demonstrates the benefit of sparse training compared to smaller dense models.

Table 1 lists accuracies for dense models and their differences for sparse models (adding the two numbers produces accuracies for sparse models) across various tasks as a function of model size $d$ (or the ratio of weights being used for the sparse models). We find using our recommendations most sparse models can be halved in size ($d = 0.5$) without sacrificing any task accuracy, whereas training with a quarter of the weights ($d = 0.25$) often reduces accuracy by less than $1\%$. Some exceptions include efficient convolutional models and sparser models ($d = 0.1$), which may be constrained by capacity.

### 5.2 COMPARISONS TO SPARSITY RESEARCH

We also draw comparisons to sparsity literature as detailed below.

**LOTTERY.** We construct lottery tickets (LOTTERY) by training models to completion (e.g., $k$ steps) and removing weights based on their trained values (Frankle & Carbin, 2019; Frankle et al., 2019). Sparse models are initialized with weights obtained after some amount of training ($t = \epsilon$) and then trained for $k - t$ steps. We choose $\epsilon \in [k/10, k/100]$ across various workloads (Frankle et al., 2019).

**SET AND RIGL.** Participating weights are rewired over time based on magnitude and adding new ones either randomly (SET) (Mocanu et al., 2018) or based on their gradients (RIGL) (Evci et al., 2020a). Weights are initialized to zero when they become participating. The fraction of weights to rewire decays linearly throughout training. We sweep across workloads for the best initial value for the rewiring fraction and frequency.

Table 1: Accuracies for dense and their differences for sparse (positive means better) for different $d$.

| Model | DENSE | $d=0.5$ | $d=0.25$ | $d=0.1$ | Model | DENSE | $d=0.5$ | $d=0.25$ | $d=0.1$ |
|---|---|---|---|---|---|---|---|---|---|
| ResNet18 | 70.30 | +0.01 | −0.92 | −2.75 | SqueezeNet V1 | 60.77 | −0.88 | −5.20 | − |
| ResNet34 | 73.87 | −0.20 | −0.65 | −2.27 | MobileNet V2 | 71.53 | −0.87 | −0.87 | − |
| ResNet50 | 76.71 | −0.05 | −0.59 | −2.17 | Stacked UNet-64 | 69.53 | −1.41 | −3.79 | −8.19 |
| ResNet101 | 77.50 | −0.25 | −0.61 | −1.52 | SSD-ResNet18 | 19.15 | −0.81 | −2.49 | −5.55 |
| ResNeXt50 | 77.68 | +0.02 | −0.49 | −2.01 | SSD-ResNet50 | 24.93 | −0.52 | −2.06 | −5.31 |
| ResNeXt101 | 79.27 | +0.20 | +0.19 | − | Faster R-CNN | 37.71 | −0.25 | −1.61 | −5.51 |
| WideResNet50 | 78.13 | +0.13 | −0.34 | −1.06 | Mask R-CNN | 38.26 | −0.34 | −1.42 | −4.80 |
| WideResNet101 | 78.63 | −0.12 | +0.04 | −1.00 | Mask R-CNN | 35.03 | −0.88 | −2.65 | −7.44 |
| InceptionV3 | 77.10 | −0.08 | −0.88 | −3.25 | Mask R-CNN 3× | 40.78 | −0.12 | −1.12 | −3.51 |
| Xception | 79.28 | +0.04 | −0.28 | −1.26 | Mask R-CNN 3× | 37.05 | −0.03 | −0.81 | −2.98 |
| DenseNet121 | 75.46 | −0.46 | −1.75 | −4.35 | RetinaNet | 36.48 | −0.42 | −2.42 | −6.00 |
| DenseNet161 | 78.77 | +0.01 | −0.86 | −2.35 | RPN | 57.61 | −0.16 | −0.90 | −2.56 |
| DenseNet169 | 76.97 | +0.04 | −0.96 | −3.23 | DETR | 39.90 | +0.10 | −0.60 | −0.80 |
| VGG11-BN | 70.70 | −0.33 | −0.79 | −2.24 | Pix2PixHD | 68.83 | +2.27 | −3.02 | −3.52 |
| VGG16-BN | 74.00 | −0.25 | −0.46 | −1.75 | Few-Shot Vid2Vid | 25.78 | +0.52 | −0.63 | −4.52 |
| VGG19-BN | 74.88 | +0.09 | −0.48 | −1.52 | FAZE | 2.94 | −0.04 | +0.02 | −0.04 |
| DRN-C-26 | 75.22 | −0.30 | −0.74 | −2.35 | Vaswani Base | 26.87 | −0.68 | −1.92 | −3.62 |
| DRN-C-42 | 76.78 | −0.10 | −0.62 | −1.98 | Vaswani Large | 28.43 | −0.09 | −0.92 | −2.12 |
| DRN-A-50 | 78.30 | −0.23 | −0.74 | −2.28 | Levenshtein | 6.16 | −0.11 | −0.23 | −0.45 |
| DeiT Tiny | 72.70 | −2.81 | −8.09 | −16.49 | GNMT | 24.81 | −0.12 | +0.26 | −0.15 |
| DeiT Small | 80.08 | −1.53 | −3.75 | −8.30 | XL Base | 22.88 | −0.49 | −2.04 | −5.41 |
| DeiT Base | 81.95 | −0.75 | − | − | XL Large | 17.90 | −0.16 | −1.01 | −2.65 |
| ShuffleNetV2 | 68.44 | −0.43 | −1.44 | − | BERT Base | 87.66 | −0.04 | − | − |
| MNASNet V1 | 71.80 | −1.09 | −3.36 | − | BERT Large | 90.92 | −0.02 | − | − |

Figure 4 plots the task error across the sparsity methods. We find that our strategy vastly outperforms competing approaches on all tasks and model sizes by a wide margin that increases with sparsity. Interestingly, the methods compared typically omit at least one of the recommendations listed in this work. For example, LOTTERY performs worse even when starting from a dense model because it does not rewire. Conversely, SET and RIGL underperform because they do not accumulate gradient updates. However, other works (Zhou et al., 2021; Hubara et al., 2021) that adopt concepts similar to those explored in this paper should perform similarly to our approach. As a result, leveraging short-term correlations seems to be a critical component for recent success in sparse training.

## 5.3 APPLICATION ON HARDWARE ACCELERATORS

We have shown earlier that we can train sparse neural models while maintaining accuracy. However, such models cannot be accelerated on modern hardware with current matrix-math pipelines (Park et al., 2017; Gale et al., 2020) without imposing particular structures (or positions of weights used during training and inference). On the other hand, neural structures targeting hardware acceleration (e.g., removing blocks (Gray et al., 2017), channels or filters (Wen et al., 2016; Li et al., 2017),

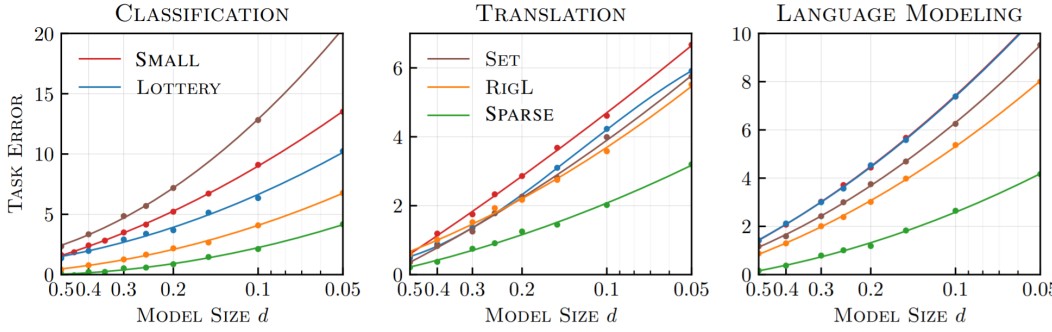

Figure 4: Accuracy difference between dense and sparse models comparing various methods as a function of model size $d$. Left to right: ResNet50 (Classification), Transformer (Translation), Transformer-XL (Language Modeling). Appendix F covers a broader span of data.

Table 2: Accuracies for dense and their differences for 2:4 sparse (positive means better).

| Model | DENSE | 2:4 1D | 2:4 2D | Model | DENSE | 2:4 1D | 2:4 2D |
|---|---|---|---|---|---|---|---|
| ResNet18 | 72.17 | +0.00 | −0.28 | VGG19 | 74.88 | +0.04 | −0.21 |
| ResNet34 | 75.14 | +0.06 | −0.27 | Xception | 79.28 | +0.04 | −0.11 |
| ResNet50 | 77.67 | +0.05 | +0.09 | DETR | 39.90 | −0.30 | −0.40 |
| ResNet101 | 78.94 | −0.09 | −0.31 | Pix2PixHD | 68.83 | +1.34 | −0.68 |
| InceptionV3 | 78.11 | −0.11 | −0.18 | Few-Shot Vid2Vid | 26.06 | +0.49 | −0.04 |
| ResNext50 | 78.36 | −0.21 | −0.19 | FAZE | 2.49 | +0.08 | − |
| ResNext101 | 79.27 | +0.28 | +0.36 | GNMT | 24.81 | +0.15 | +0.09 |
| WideResNet50 | 78.13 | −0.07 | −0.08 | Transformer Large | 28.43 | −0.10 | −0.39 |
| WideResNet101 | 78.63 | +0.08 | −0.07 | BERT Large | 90.92 | +0.03 | −0.55 |
| DRN C 26 | 77.66 | −0.05 | −0.11 | | | | |

layers (Michel et al., 2019)) often degrade accuracy. This apparent tradeoff between accuracy and performance (or speed) has hindered their adoption.

Therefore, we explore whether our recommendations can make hardware-aware structures more amenable for deep learning. As a case study, we consider Sparse Tensor Cores introduced in NVIDIA Ampere GPU architecture (Mishra et al., 2021), which have twice the math throughput of regular matrix operations. The hardware expects a 2:4 sparsity structure that takes at least two values to be zero for each group of four values. Appendix E illustrates examples of 2:4 sparsity for inference (1D) and for training (2D).

Table 2 lists accuracy differences between sparse and dense models (positive values mean sparse performs better), where we extend learning rate schedules for some workloads compared to what is typically used in literature (see Appendix D for details). From the table, we find that structured sparse neural models can generally retain accuracy for tasks. While neural models adopting coarser structures such as block sparsity (Narang et al., 2017; Gray et al., 2017) fail to benefit from our recommendations and perform no better than smaller models (see Appendix E), using 2:4 for training (2D) and inference (1D) roughly matches accuracy of the dense models. Therefore, we can conclude 2:4 sparsity is particularly effective at leveraging short-term correlations due to its finer granularity. This suggests possible avenues towards accelerating training (Hubara et al., 2021) as well as obtaining efficient models for inference without having to repeat the training process (Mishra et al., 2021; Zhou et al., 2021).

## 6 RELATED WORK

Applying sparsity to reduce the size of neural models has been a topic of interest for the past three decades (LeCun et al., 1990; Hassibi & Stork, 1993; Reed, 1993; Castellano et al., 1997; Han et al., 2015; Jayakumar et al., 2020). Typically, sparse models are constructed by training much larger models (that are easier to train) and removing some of their weights either after training (Han et al., 2015; Mishra et al., 2021) or gradually alongside training (Narang et al., 2017; Zhu & Gupta, 2018; Gale et al., 2019). However, the above are only useful to reduce costs for inference.

For training acceleration, early works (Lee et al., 2019; Wang et al., 2020; Tanaka et al., 2020) sought to remove weights before training with limited success (Gale et al., 2019; Frankle et al., 2021). Surprisingly, it was shown sparse models can be trained when initialized the same way as the trained dense model (Frankle & Carbin, 2019). After this breakthrough, many works (Dettmers & Zettlemoyer, 2019; Evci et al., 2020a; Jayakumar et al., 2020) tried to dynamically learn sparse models by rewiring their weights throughout training, though some works existed earlier (Bellec et al., 2018; Mocanu et al., 2018). While this greatly improved accuracy, it was not enough to match that of dense models. More recent advances (Wortsman et al., 2019; Liu et al., 2020; Savarese et al., 2020; Zhou et al., 2021; Hubara et al., 2021) introduced gradient updates for all weights in a neural model, including ones that do not participate in training, and observed better success.

Our recommendations are similar to recent works (Zhou et al., 2021; Hubara et al., 2021) that were conducted in tandem with our research, but differ in some crucial aspects. (Zhou et al., 2021) adopts

2:4 1D sparsity patterns which cannot accelerate training. We also show alternatives to regularizing the non-participating weights which achieve the same effect. (Hubara et al., 2021) uses 4:8 sparsity patterns which are not useful for Sparse Tensor Cores, but they also adopt 2D patterns as we do.

While sparsity has been extensively explored for neural model training, our paper presents new perspectives on why certain methods should or not work. Our explanations about how additional weights do not learn meaningful information throughout training complement existing observations about the benefits of overparameterization (Evci et al., 2020b). We use this understanding to consolidate many ideas that have been circling around in sparsity literature but remain not clearly understood. For example, how frequently and why must weights be rewired.

## 7  CONCLUSION

In this paper we provide a better understanding of what is needed for sparse training. We first discover weights that can be pruned during inference do not learn meaningful representations over time, but exhibit short-term correlations that are necessary to train effectively. Based on this understanding, we uncover recommendations to remove such weights when training sparse models and conduct extensive empirical studies to determine the effects of different hyperparameters. We then demonstrate that sparse training can work across dozens of deep learning workloads that have not been investigated before in sparsity literature. Lastly, we are the first to employ sparsity patterns that can accelerate training of sparse models on existing hardware.

We believe these results open many questions. On the practical side, it may be interesting to consider how could this strategy be adopted to accelerate real-world training and inference workloads today, reducing the environmental impact from training very large models. On the theoretical side, we would like to understand better ways to approximate the short-term behavior in cases where accuracy still suffers as well as how to address potential biases induced by sparse models which may affect applications. We hope that our results will spur further research on these unconventional architectures, which challenge the default choice held by massive neural models today.

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

## A    RECOMMENDATIONS FOR TRAINING

We expand our investigations on how to remove weights that do not learn meaningful information when training sparse models, covering more neural architectures and deep learning tasks.

Figure 5 plots the task error as a function of rewiring steps $r$ for sparse models of different $d$. We observe that error increases with less frequent rewiring ($r \to \infty$) for other vision tasks, since rewiring is related to how often traversal through the parameter space expands.

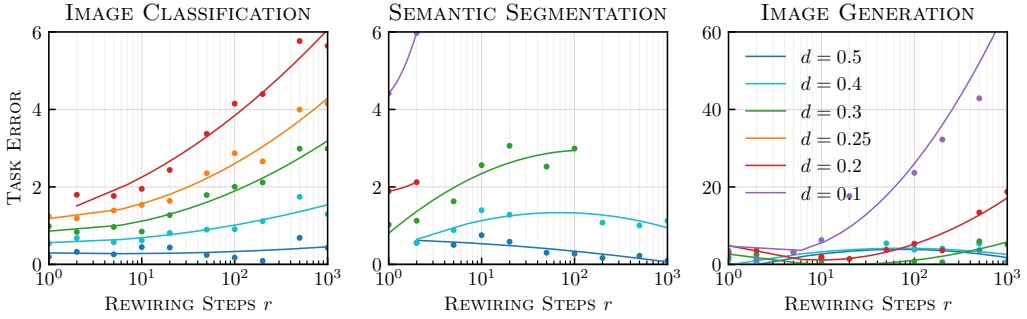

Figure 5: Same as Figure 2 (left). From left to right: InceptionV3 (Image Classification), Mask RCNN (Semantic Segmentation), Pix2PixHD (Image Generation).

Figure 6 shows the task error as a function of the scale factor $s$ applied to gradient updates for non-participating weights. We observe the error increases with decreasing contributions of the gradients ($s \to 0$), which suggests updates to non-participating weights are also important during training for other tasks.

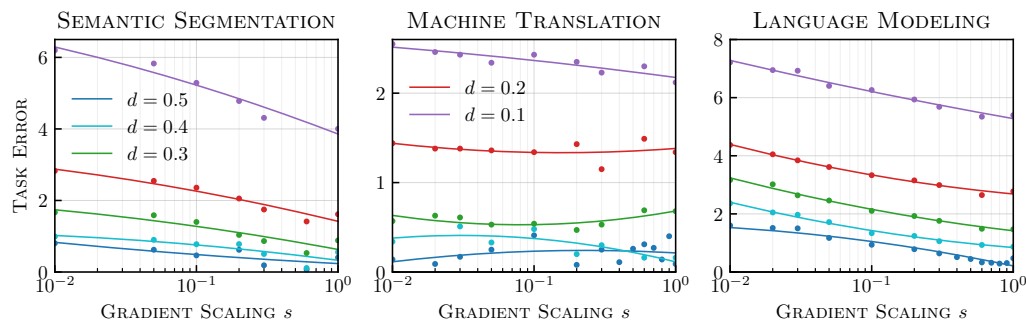

Figure 6: Same as Figure 2 (middle). From left to right: Mask R-CNN (Semantic Segmentation), Transformer (Machine Translation), Transformer-XL (Language Modeling).

Figure 7 demonstrates the task error as a function of the number of training steps $z$ at which non-participating weights are reset to zero. Similar to results in Section 3.2, error rates saturate after sufficient training ($z \sim 1k$), which reinforces the idea that non-participating weights augment parameter spaces rather than model capacity.

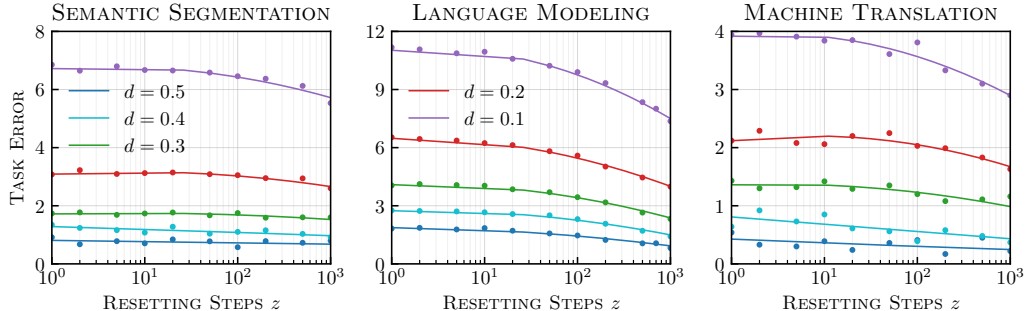

Figure 7: Same as Figure 2 (right). From left to right: Mask RCNN (Semantic Segmentation), Transformer-XL (Language Modeling), GNMT (Machine Translation).

Figure 8 compares various exploration and exploitation strategies for training sparse models, as described in Section 3.3. While task accuracy degrades with lack of proper exploration or exploitation, inducing exploitation by removing gradient noise from non-participating weights (FIX, RESET, REGULARIZE) substantially decreases the error rates.

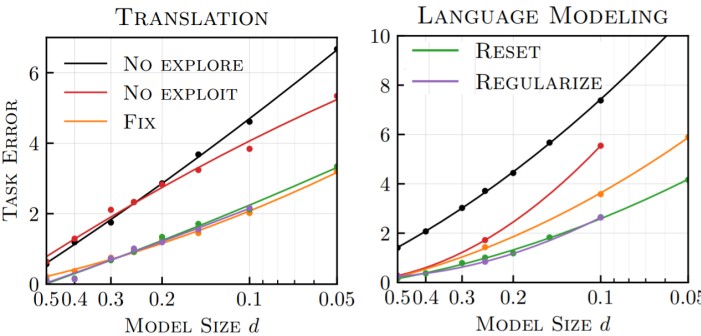

Figure 8: Same as Figure 3. From left to right: Transformer (Translation), Transformer-XL Large (Language Modeling).

The above results across more deep learning workloads further validate recommendations put forth in Section 3 for training sparse models.

## B    EXPERIMENTAL SETUP

We design experiments on PyTorch (Paszke et al., 2019) using custom autograd functions for convolutions and linear layers. Our functions emulate sparsity using a binary tensor (or mask) that we multiply elementwise with the weights of each layer during forward and backward propagation. We determine masks based on weight magnitudes, as described earlier, after the optimizer step and before the next training iteration.

### B.1    IMAGE CLASSIFICATION

We train popular convolutional models like ResNets (He et al., 2016), VGG (Simonyan & Zisserman, 2015), Stacked U-Nets (Shah et al., 2018), Dilated Residual Networks (Yu et al., 2017), Inception (Szegedy et al., 2015), MobileNet (Sandler et al., 2018), as well as vision transformers like DeiT (Touvron et al., 2020). Training involves standard pipelines for image classification on ImageNet-2012 as described in literature and found in public code repositories. For most workloads, we adopt learning rates with linear warmups for the first 5 epochs, drop the learning rate by a factor of ten at epochs 30-60-80, and stop training after 90 epochs. A few select neural models (e.g.,

mobilenets and vision transformers), however, are trained for more epochs using linear or cosine schedules.

We measure model quality using top-1 classification accuracy. We apply sparsity to convolutions and linear layers with some exceptions: convolutions whose input channels are not divisible by 16 (e.g., first convolution layer, group and depthwise separable convolutions).

### B.2 IMAGE SEGMENTATION AND DETECTION

Image segmentation and detection tasks include both regression and classification components. Popular detectors and segmentors are typically trained in two phases: first a backbone is trained for image classification, followed by the addition of model components that are trained for detection or segmentation. Backbones are trained on ImageNet-2012, while downstream tasks are trained on COCO. We adapt training scripts and code from Detectron2 (Wu et al., 2019).

We train neural models such as regions with convolutional neural networks (R-CNN) variants (Ren et al., 2015; He et al., 2017), vanilla one-shot detectors (Liu et al., 2015), and with focal loss (Lin et al., 2017). Convolution and linear layers encountered in pretrained backbones are sparse, like for classification tasks. Detection and segmentation heads are also targeted.

### B.3 GENERATIVE MODELING

Generative Adversarial Networks (GANs) contain two subnetworks: a generative model and a discriminative model which combine regression and discrimination tasks during training. For image and video tasks, the generator model regresses pixel colors. We explore conditional GANs for super image-to-image translation and video-to-video synthesis using Imaginaire (NVIDIA, 2020b). We measure quality of generated outputs using the Frechet Inception Distance (FID). We experiment with generative neural models like Pix2PixHD (Wang et al., 2018b), Vid2Vid (Wang et al., 2018a), and FewShot-Vid2Vid (Wang et al., 2019), targeting convolution and linear layers.

### B.4 MACHINE TRANSLATION

We explore transformer and recurrent neural models for language translation. All models are encoder-decoder style architectures trained for English to German (En-De) translation on WMT. We adapt model and training code from Fairseq (Ott et al., 2019) and NVIDIA Deep Learning Examples (NVIDIA, 2020a). We measure model quality using BLEU scores.

We experiment with GNMT (Wu et al., 2016) and transformer-based architectures (Vaswani et al., 2017; Gu et al., 2019). All linear layers are sparse, except for embeddings and vocabulary projections.

### B.5 LANGUAGE MODELING

We consider recent advances in word-level language modeling using transformer decoder (left-to-right) or encoder (bi-directional) architectures (Radford et al., 2018; Devlin et al., 2018). We pretrain language models in an unsupervised fashion on WikiText-103 or Wikipedia corpus, and evaluate on downstream tasks that are zero shot or require additional finetuning. We train them using Megatron (NVIDIA, 2020c) and NVIDIA Deep Learning Examples (NVIDIA, 2020a). Model quality is measured in terms of perplexity or F1 score.

We train language models such as Transformer-XL Dai et al. (2019) and BERT Devlin et al. (2018). We make all linear layers sparse, except for embeddings, vocabulary projections, and classification heads for downstream tasks.

## C COMPARISONS TO DENSE MODELS

It is also interesting to understand where does accuracy of our sparse models (SPARSE) fall between accuracies of dense (DENSE) and smaller neural models (SMALL). We define a metric $f = (\text{DENSE} - \text{SEARCH})/(\text{DENSE} - \text{SMALL})$, where a value of one means accuracy matches that of dense, and zero implies accuracy is no better than a smaller model.

Figure 9 illustrates $f$ as a function of model size $d$ for various tasks. For $d \sim 0.5$, sparse models are able to approximate dense models across all neural architectures and tasks. While $f$ decreases for sparser models ($d \ll 0.5$), they are still much more efficient than smaller models. Even at $d = 0.1$, most models can retain a large amount of accuracy.

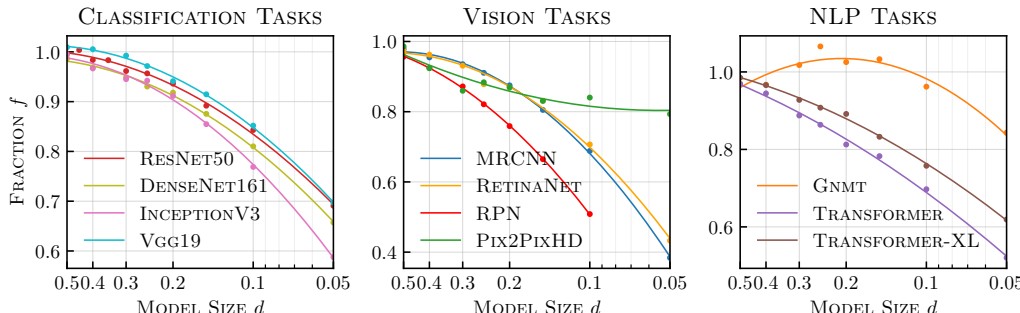

Figure 9: Fraction of accuracy that sparse models achieve between dense and smaller models as a function of $d$ across various tasks and neural architectures. We measure the fraction as $f = (\text{DENSE} - \text{SEARCH})/(\text{DENSE} - \text{SMALL})$.

## D    EFFECTS OF LONGER TRAINING

Since conventional models can achieve higher accuracy when trained on larger data sets or with longer training schedules (Liu et al., 2019), another interesting direction is to explore the effects of sparsity on models that are trained to the limits of their capacity. We train neural models longer by extending their learning rate schedules after warmup , e.g. a schedule $1/t$ becomes $2/t$.

Figure 10 (left) plots accuracy differences as a function of training time $t$, which denotes the ratio of training steps to the original training schedule. The fact accuracy improves with more training suggests neural models are often not trained to capacity using conventional schedules. We observe sparse models achieve worse accuracy than dense models that are undertrained ($t \sim 1$), but can match accuracy when trained to capacity ($t \gg 1$).

Figure 10 (middle) illustrates task errors between dense and sparse models that have been trained for the same number of steps. We find error scales inversely with $t$, since more training gives neural models better chances to explore parameter spaces. Particularly, errors can reach zero with sufficient training (Evci et al., 2020a) for models that are not constrained by capacity ($d \geq 0.25$). Figure 10 (right) shows the time it takes for sparse models to recover accuracy of dense models is relatively short when $t \sim 1$, but significantly longer as accuracy saturates ($t \gg 1$).

Investigations on the effects of longer training are expanded to more neural architectures and deep learning tasks. Figure 11 illustrates accuracy deltas as a function of training time $t$. For vision tasks, we find sparse models of moderate sizes ($d \geq 0.25$) can match accuracy of dense models after sufficient training ($t \geq 3$). On the other hand, for language modeling, sparse models cannot match accuracy for any time $t$ because dense models are already near capacity for the task at hand. Obviously, when using popular training schedules ($t \sim 1$), sparse models can be trained a bit longer to recover the accuracy lost.

## E    APPLICATION ON HARDWARE ACCELERATORS

This appendix explores various sparsity structures considered in the paper that are amenable for acceleration using modern matrix-math hardware.

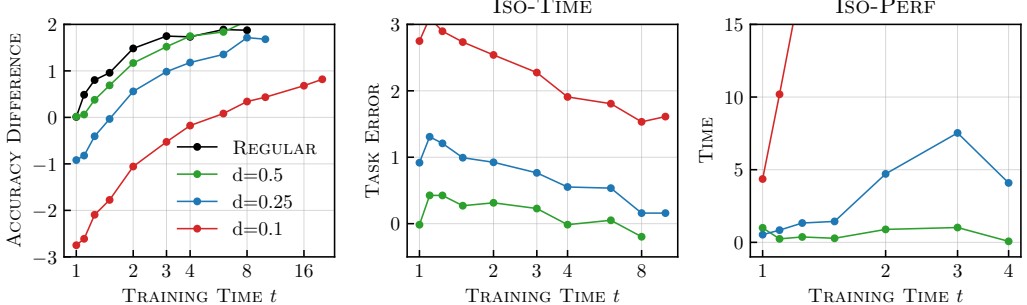

Figure 10: Effects of sparsity with longer training of ResNet18. Left: Accuracy differences as a function of training time $t$ (or ratio of steps to the original training schedule). Middle: Task error between dense and sparse models trained for the same duration. Right: Training time it takes for sparse models to match accuracy of dense models.

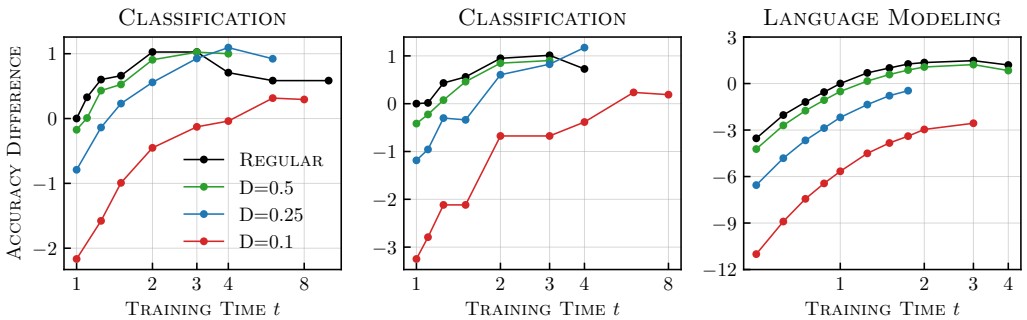

Figure 11: Same as Figure 10 (left). Left to right: ResNet50 (Classification), InceptionV3 (Classification), Transformer-XL (Language Modeling).

### E.1 BLOCK SPARSITY

We first consider block sparsity (Narang et al., 2017; Gray et al., 2017), which removes blocks of contiguous elements (or weights) in a neural layer as shown in Figure 12. Block sparsity addresses common issues that are present for unstructured formats: indices for active blocks reduce storage overhead by a factor of the block size, blocks are stored contiguously in memory which reduces irregular memory accesses, and their computations can exploit faster matrix-math hardware, such as Tensor Cores in NVIDIA GPUs.

We construct block sparse structures by (1) partitioning a neural layer into a set of blocks, (2) aggregating elements in each block into a metric, and (3) removing blocks according to some criteria based on their metrics. While we remove blocks based on largest magnitude $\max_{i\in(1,b^2)} w_i$. Other choices such as the $p$-norm $(\sum_i^{b^2} |w_i|^p)^{1/p}$ achieve similar results.

Because structures restrict the combination of weights that can be formed in a neural model, an important question is then for what block sizes $b$ (if any) can sparse models retain accuracy. Table 3 lists accuracy differences between sparse and dense models for $d = 0.5$ using various block sizes. We find block sparse models fail to maintain accuracy, performing no better than smaller models. Notably, accuracy deteriorates for all tasks and block sizes, including smaller blocks ($b \leq 4$) that are less amenable for hardware acceleration. In other words, block structures are too coarse for retaining short-term correlations. For example, different weights in a block may have different roles during training: a block that participates in training may contain weights that are not important, wheres a non-participating block may have weights that were crucial to keep. Both cases prevent sparse

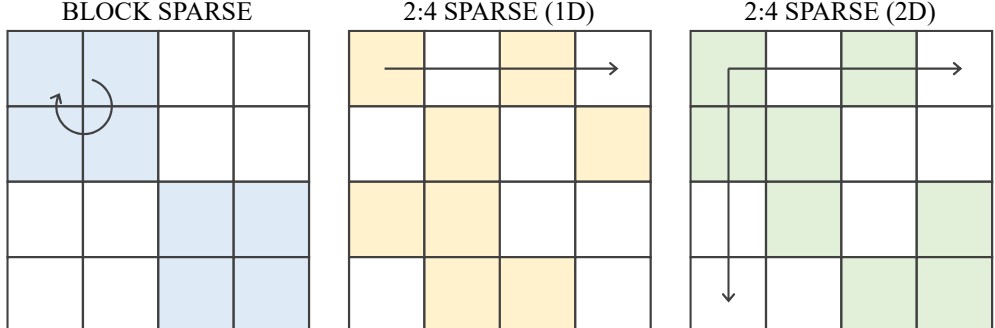

Figure 12: Block sparse, 2:4 1D, and 2:4 2D structures for a $4\times4$ neural layer. Blank cells represent weights that do not participate in training (are assumed zero) and colored cells denote weights that participate (have nonzero values). Arrows indicate direction along which structure is imposed.

Table 3: Accuracies for dense models and their differences for block sparse models using different block sizes $b$.

| Model | DENSE | SMALL | $b=1$ | $b=2$ | $b=4$ | $b=8$ | $b=16$ | $b=32$ |
|---|---|---|---|---|---|---|---|---|
| Transformer-XL | 22.88 | $-2.48$ | $-0.82$ | $-2.03$ | $-2.35$ | $-2.24$ | $-2.36$ | $-2.41$ |
| Transformer | 28.43 | $-0.77$ | $-0.21$ | $-0.75$ | $-1.10$ | $-1.47$ | $-1.98$ | $-1.94$ |
| GNMT | 24.81 | $-2.75$ | $-0.15$ | $-0.15$ | $+0.02$ | $-0.08$ | $-0.15$ | $+0.30$ |
| ResNet50 | 76.71 | $-1.63$ | $-0.46$ | $-0.99$ | $-2.19$ | $-2.79$ | $-$ | $-$ |
| Mask RCNN | 35.03 | $-0.88$ | $-0.28$ | $-1.17$ | $-2.43$ | $-3.74$ | $-4.41$ | $-5.00$ |

models from retaining weights over time that are relevant for traversing the parameter space, and thus impacting accuracy.

### E.2    2:4 SPARSITY

We next consider Sparse Tensor Cores (Mishra et al., 2021) introduced in NVIDIA Ampere GPU architecture which exploit 2:4 sparsity and have twice the math throughput of regular matrix units. Figure 12 shows that 2:4 sparsity mandates each group of four values must have at least two values that are zero. Typically, 2:4 is applied on weights $w$ in the forward pass, $y = wx$. However, we can also apply 2:4 on weight transposes $w^T$ for the backward pass, $\partial L/\partial x = \partial L/\partial y \times w^T$. We denote these two options as 2:4 1D that accelerates forward pass for inference Mishra et al. (2021), and 2:4 2D that accelerates both forward and backward passes for training.

The 2:4 sparsity structure must always be imposed along the inner dimension of dot products. For linear layers, we apply 2:4 on a $n \times k$ weight tensor along $k$ or $n$ (for forward or backward pass, respectively). For convolutions, we apply 2:4 on a $k \times c \times r \times s$ weight tensor along input channels $c$ or $k \times r \times s$ (for forward or backward pass, respectively), where $k$ denotes output channels, $r$ and $s$ are kernel dimensions.

We can satisfy 2:4 1D constraints by removing weights with lowest magnitudes. Since 2:4 2D constraints have no trivial solution, we seek to minimize the cumulative magnitude of the removed weights. In other words, for each $4 \times 4$ block in the tensor, we construct all possible combinations of 2:4 2D patterns, compute their 1-norm, and choose the structure that has the largest norm.

## F    COMPARISONS TO SPARSITY RESEARCH

We expand our comparisons to literature covering more neural architectures and deep learning tasks. Figure 13 illustrates the task error across various methods, neural architectures, and tasks. We find our strategy outperforms competing approaches in most cases with a few exceptions: we do not

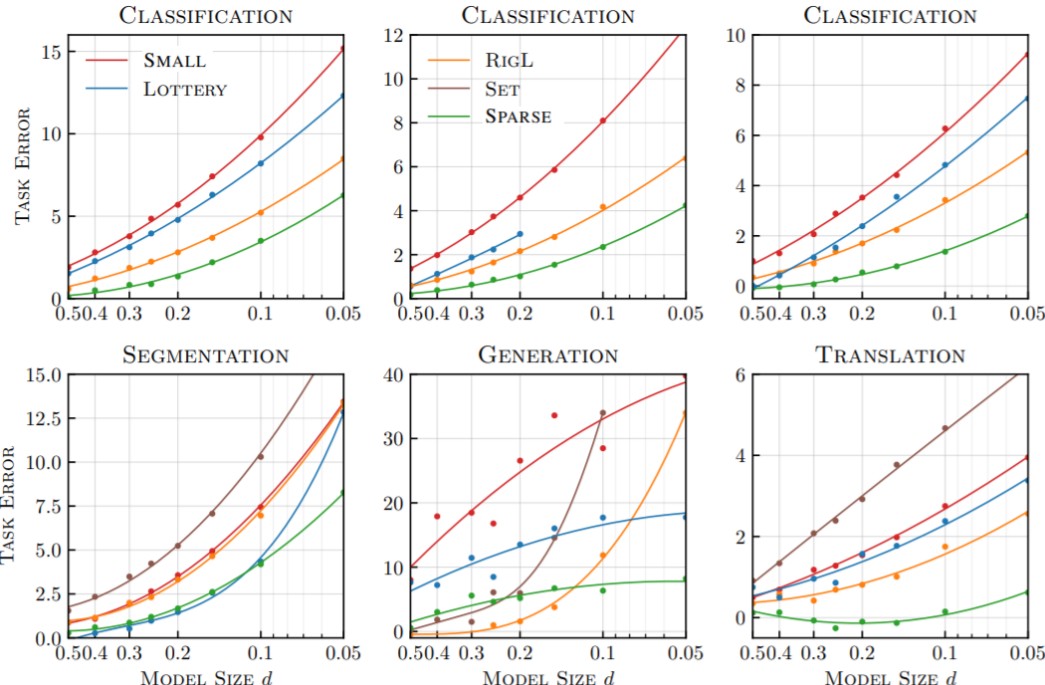

Figure 13: Same as Figure 4. Clockwise: InceptionV3 (Classification), DenseNet161 (Classification), VGG19 (Classification), GNMT (Translation), Pix2PixHD (Generation), Mask RCNN (Segmentation).

outperform lottery tickets in segmentation tasks because detectors and segmentors are trained with small learning rates which limit exploration. For generation tasks, our models are slightly worse for $d \sim 0.5$ (though noisy scores make it difficult to draw conclusions), but the asymptotic behavior of error rates as $d \rightarrow 0$ clearly indicates that our approach is superior.

