# OpenReview forum: "Search Spaces for Neural Model Training"
_ICLR.cc/2022/Conference — ICLR 2022 Submitted_

### Official Review · Reviewer_yWXp · 2021-10-31

**Correctness:** 4
**Technical Novelty And Significance:** 4
**Empirical Novelty And Significance:** 3
**Recommendation:** 6
**Confidence:** 4

**Main Review:**

Strengths:
1.	It is interesting to understanding the training of deep models from the perspective of search space.
2.	This paper proposed a series of steps to train the sparse models as larger models, and quantitative experimental results show that these steps can effectively reduce the training error of the sparse model

Weaknesses:

1. The third paragraph of Section 2.1 introduces the changes in the role played by model weights during the training process (shown in Fig. 1). And authors make an assumption that the weights which are eventually discarded (or added weights) play a crucial role in determining what will be used for inference. Nevertheless, it is hard to verify the role of these weights according to existing results. More results and explanations are required.

2. Since this paper focuses on training sparse models, it would be better to compare the proposed method with model compression methods (e.g., [a,b]).

3. The authors propose several techniques to encourage the exploration and exploitation when training deep models. However, it is still unclear how to derive the final sparse model from a full network. More details should be provided.


Reference:

[a] Discrimination-aware network pruning for deep model compression. PAMI 2021.

[b] ResRep: Lossless CNN Pruning via Decoupling Remembering and Forgetting. ICCV 2021.


**Summary Of The Paper:**

The authors proposed a new point of view to explain why deep learning models often require more weights are needed to train models rather than to run inference for tasks. One example is that sparse models usually perform well in inference, but do not perform well in training. The main content of this viewpoint is that more weights during training can expand the search space, create extra degrees of freedom for search, and form new optimization paths when model optimization falls into critical points, thereby facilitating model training. Based on this point of view, this paper proposes a series of steps to augment search spaces of sparse models during training to approximate the behavior of larger models. The experimental results show that this series of steps can effectively reduce the model error.

**Summary Of The Review:**

It is an interesting paper and the experiments are strong for me.

---

> ### Author Response · Authors · 2021-11-23
> **Paper4028 Author Responses to Questions from Reviewer yWXp (Part 1/1)**
>
> We thank the reviewer for their time and careful consideration. We address the reviewer's question below.
>
> **The third paragraph of Section 2.1 introduces the changes in the role played by model weights during the training process (shown in Fig. 1). And authors make an assumption that the weights which are eventually discarded (or added weights) play a crucial role in determining what will be used for inference. Nevertheless, it is hard to verify the role of these weights according to existing results. More results and explanations are required.**
>
> We have made significant changes to Section 2 in the paper to address concerns by several reviewers on its novelty in relation to other works. Since we base our claims on findings from relevant literature, we believe that most of these claims are well-founded. We direct the reviewer to our response to Reviewer kChj.
>
> **Since this paper focuses on training sparse models, it would be better to compare the proposed method with model compression methods (e.g., [a,b]).**
>
> We agree that such comparisons could be made. However, the main focus of this paper is on accelerating training, so we avoid making many comparisons to methods that are designed for inference, of which there are many.
>
> **The authors propose several techniques to encourage the exploration and exploitation when training deep models. However, it is still unclear how to derive the final sparse model from a full network. More details should be provided.**
>
> The final sparse model is obtained by keeping the weights that participate in the last training iteration and pruning away those that have been pruned and do not participate. We moved the algorithm from the Appendix to Section 4 to better explain how sparse models are generated.

---

### Official Review · Reviewer_t5MH · 2021-11-02

**Correctness:** 3
**Technical Novelty And Significance:** 2
**Empirical Novelty And Significance:** 3
**Recommendation:** 6
**Confidence:** 4

**Main Review:**

This work has two main contributions. First authors argue that over-parameterization is useful for `search`. Later authors provide extensive experimental results (maybe the largest I've seen in this field) on comparing various sparse training techniques. I thank authors for their extensive experimental results, which is quite useful to the community, as many researcher can't do such extensive experiments. This contribution alone could have a significant impact on the field. Do they plan open-sourcing their code? The idea of grouping weights into 3 subsets (inactive/active/search-space) is also interesting and novel (though this idea is mentioned in various work). Though this work has quite a bit potential to impact and inspire future research, I found few things confusing and have questions about its novelty.

## Concerns
0. Main method used in the paper reminds me previous work like Discovering Neural Wirings and TopKast, in which forward pass is made on top-magnitude weights and gradients are applied to all weights. It would be nice to make a clear comparison with previous work; i.e. what is the novelty?
1. Naming in Figure-4 is a bit confusing, I recommend using terms like 'Small-Dense', 'Static' instead of no-explore/exploit (also NO EXPLORE becomes REDUCE in the next figure, which is not consistent) as it is used in previous work. Furthermore 'Fix' and 'Reset' have similar curves. So resetting doesn't help? Then, the text needs to be updated as it claims 'fix' brings better results. Maybe bringing results from Figure:12 that shows the gains of resetting weights might be useful.
2. rigl is known to take longer training steps to converge. Are your experiments made with default dense training steps? I recommend authors to do extended training comparisons (like 2x or 5x) for the Resnet50 classification experiments. Similarly do you use the ITOP recipe proposed by [4] in your experiments? It would be nice to make this clear in the text, as if using limited training steps, ITOP improves the performance of rigl. And finally the difference between 'rigl/set' and 'search' needs to be clear in the text. fully-sparse DST methods (set/rigl) don't need to store values of the inactive weights, which reduces the peak memory consumption significantly. I recommend authors to make this distinction early in the paper, motivate the setting they are focusing on and clearly indicate before each experiment.
3. I think part of Appendix C should be brought to the main text (like the algorithm), as it would help the reader to understand the experiments and contributions better. One way to enable that would be to make Section 2 more succinct, as few ideas are repeat in different subsections.
4. Section 2.4 seems very similar to the experiments/observation in [3] (and some other parts of Section 2). Authors should discuss similarities and differences. I am not sure what new insights are provided in Section 2 compared to [3]. Can you make this clear?
5. Hypothesis and evidence provided in section 2.3 is not convincing to me. When weights approach to zero, they naturally become uncorrelated. Therefore the correlation tells more about which weights go to zero than anything else I think. Furthermore since low magnitude weights are selected at the end it becomes a bit circular. Finally, the relationship between correlation and exploration is not clear.
6. In appendix it says "For smaller models, we reduce the width of neural layers by a factor of 1/d to match the number of weights used in sparse models." The factor should be 1/sqrt(d), as width=0.1, would have roughly 1% of the parameters.
7. weight decay has an important effect on exploration and used successfully in literature [1,2]. Do you use weight decay in your experiments?  Are they included in the gradients when you scale the them for Figure:3?
8. I am not sure the illustration on Figure-2 is helpful giving an example where extra dimensions can be helpful. Since even with the vertical dimension, the optimization would stay at the bad local minima for the given example. Maybe "mexican-hat" example would be a better fit.

## Minor
- First paragraph of Section 2.1, authors should cite the original work on each point made as much as possible. Happy to suggest early references if needed.
- "since they contribute the most" -> the least
- "They are also are tolerant"
- "...use search spaces to describe reasons more weights are needed for training" - "to describe why more weights are needed..."
- "Task error" doesn't seem like the right term. I would recommend using something like 'Delta Accuracy' or 'Compression Error'
- In captions ResNet50 is mentioned without the dataset. I assume it is Imagenet?
- "by training sparse models delineated so far" Not clear what this refers to. So far when?
- RIGL (Mocanu et al., 2018) (typo)
- Figure 17: Clockwise is ambiguous. update titles with model names?
- Related Work/ 'which improved accuracy but not enough to match that of larger models' This statement is vague and possibly not correct. This depends on the sparsity. Maybe you mean dense-to-sparse training? Then, rigl and topkast do match pruning results.
- 'these results open many question'

[1] https://arxiv.org/abs/2110.00296
[2] TopKAST: https://proceedings.neurips.cc/paper/2020/file/ee76626ee11ada502d5dbf1fb5aae4d2-Paper.pdf
[3] https://arxiv.org/abs/1906.10732
[4] https://arxiv.org/abs/2102.02887

## After Rebuttal
I read author's response and looked the revised version quickly. A lot seems to be changed during rebuttal, which is probably a good thing; though makes it difficult to evaluate. Regardless, as I mentioned above, the experimental results (which includes basically everything except RL) is unique and thus important for informing future research, thus I raised my score to 6.

If rejected, venues that accept longer papers (like JMLR) might be a good fit. It would be great to discuss experiments in different domains in detail and try to understand/quantify why different behaviours are observed in different domains.

**Summary Of The Paper:**

This paper studies the training of sparse neural networks and the effect of extra parameters (dense connectivity) during training. They hypothesize that pruned connection are useful, because they help optimization to take some random steps in those direction which help optimization escaping the local minima. Authors support their claims with experiments and finally share a very long list of experimental results on training sparse networks.

**Summary Of The Review:**

As mentioned above the experimental results provided at the end of the paper (one of the most extensive set of experiments I have seen in a paper), would be a strong contribution alone. However, authors use half of the paper focusing on explaining why more connections are needed for better results, which I find not very novel and a bit confusing. Authors say that pruned connections are used in training for exploration and search very vaguely; which doesn't provide any new insights on the existing understanding of the training of sparse neural networks. I am willing to increase my score if my concerns are addressed and looking forward to read the next version of this work.

---

> ### Author Response · Authors · 2021-11-23
> **Paper4028 Author Responses to Questions from Reviewer t5MH (Part 1/3)**
>
> We thank the reviewer for their positive feedback. We are glad that the reviewer found our experimental results useful, as that was one of main motivations for publishing this work. Below we have addressed the reviewer's concerns.
>
> **It would be nice to make a clear comparison with previous work; i.e. what is the novelty?**
>
> As the novelty of our work in comparison to others has been questioned by several other reviewers, we made significant changes in our paper to better define these. We also included a summary of the novelty of our work in the response to Reviewer kChj. These are now more clearly defined in the abstract, introduction, and conclusion of the revised paper.
>
> **Naming in Figure-4 is a bit confusing, I recommend using terms like 'Small-Dense', 'Static' instead of no-explore/exploit (also NO EXPLORE becomes REDUCE in the next figure, which is not consistent) as it is used in previous work.**
>
> No-explore can still be using dynamic sparsity, but without the gradient updates. Similarly, no-exploit could be using dynamic sparsity and updating gradients for all weights, but does nothing to hold back the non-participating weights. Small-Dense and Static would be incorrect characterizations of these scenarios. As a result, we believe no-explore and exploit are more accurate descriptions of what we are trying to show in that subsection, even though they are not standard terminology in literature.
>
> **Furthermore 'Fix' and 'Reset' have similar curves. So resetting doesn't help? Maybe bringing results from Figure:12 that shows the gains of resetting weights might be useful.**
>
> Our belief is that ‘Fix’, ‘Reset’, and ‘Regularize’ all achieve the same effect of minimizing the noise of gradient accumulations of pruned weights. While they all achieve similar accuracies, ‘Reset’ is easier to use as the hyperparameter is more readily understood as the interval after which correlations vanish. ‘Fix’ requires knowing when gradient updates are no longer needed to explore parameter spaces, which from experience can vary between networks. ‘Regularize’ has a similar effect as ‘Reset’. We replaced the current Figure which uses ResNet50 with one of the plots in Figure 12 that uses Transformer-XL, since the comparisons are easier to make.
>
> **rigl is known to take longer training steps to converge. Are your experiments made with default dense training steps? I recommend authors to do extended training comparisons (like 2x or 5x) for the Resnet50 classification experiments.**
>
> Figure 6 shows comparisons for longer training steps, where we increase the training time of both dense and sparse models.
>
> **And finally the difference between 'rigl/set' and 'search' needs to be clear in the text. fully-sparse DST methods (set/rigl) don't need to store values of the inactive weights, which reduces the peak memory consumption significantly. I recommend authors to make this distinction early in the paper, motivate the setting they are focusing on and clearly indicate before each experiment.**
>
> We moved discussions on limitations of our approach (such as memory consumption) from the Appendix to Section 4.
>
> **I think part of Appendix C should be brought to the main text (like the algorithm), as it would help the reader to understand the experiments and contributions better.**
>
> We agree with the reviewer and have brought Appendix C to the main text in Section 4.
>
> **One way to enable that would be to make Section 2 more succinct, as few ideas are repeat in different subsections. Section 2.4 seems very similar to the experiments/observation in [3] (and some other parts of Section 2). Authors should discuss similarities and differences. I am not sure what new insights are provided in Section 2 compared to [3]. Can you make this clear?**
>
> We have revised Section 2 to reduce information already presented in [3] and emphasize on the novelty in our paper.

---

> > ### Author Response · Authors · 2021-11-23
> > **Paper4028 Author Responses to Questions from Reviewer t5MH (Part 2/3)**
> >
> > **Hypothesis and evidence provided in section 2.3 is not convincing to me. When weights approach to zero, they naturally become uncorrelated. Therefore the correlation tells more about which weights go to zero than anything else I think.**
> >
> > We believe correlations should be insensitive to the fact that weights go to zero for a few reasons.
> >
> > 1. Correlations are computed between two different time windows separated by some time delay, where each window computes the weight magnitude’s difference from the mean and normalizes with variance. As a result, the absolute weight magnitudes (i.e. whether weights are generally large or small) will not impact the correlation metric, and only their relative difference to the mean within a window should have a contribution. In other words, the expectation should be that each window is normalized (to zero mean and unit variance) and thus different windows are compared at the same scale.
> >
> > 2. Since these time windows are much smaller than the total training duration, the effect from learning rate schedules (that diminish weight updates and thus how much weights change over time) should also be minimal.
> >
> > 3. We aggregate the correlation across various time windows to compute the total correlation. Weights that move towards zero and are pruned at the end of training could in principle be correlated if they had recurring (albeit small because of low learning rates) gradient updates along the same direction. The fact that they do not correlate, means gradient updates are moving the weights along random directions. It is also important to note that weights that become small at the end of training are not necessarily small throughout all of training - a profile of their trajectories will show very unpredictable movements. Since correlations are computed throughout all of training, they will likely capture both low and high weight magnitudes.
> >
> > **Furthermore since low magnitude weights are selected at the end it becomes a bit circular.**
> >
> > We select weights based on their final magnitudes just as a means to classify large weights that are important for inference versus small weights that are not important and can be pruned. This is a typical approach used in pruning literature. We then show that correlations for important weights are much longer than those for unimportant ones. Naturally, in a real training scenario, we would not have information of which weights will become low/high at the end of training, and approximate by choosing weights based on their magnitudes at every iteration.
> >
> > **Finally, the relationship between correlation and exploration is not clear.**
> >
> > Correlations are used to measure the interval during which weights of various magnitudes learn meaningful representations. We find small weights that are pruned for inference tend to learn short-term representations that last shorter than one epoch. This means we can discard those weight values after every epoch, which is necessary to reduce the impacts of noise from gradient accumulations of inactive weights (remember that we update weight gradients for all weights, not just active weights).
> >
> > **In appendix it says "For smaller models, we reduce the width of neural layers by a factor of 1/d to match the number of weights used in sparse models." The factor should be 1/sqrt(d), as width=0.1, would have roughly 1% of the parameters.**
> >
> > By “width of neural layers” we were referring to only one of the dimensions of a weight matrix rather than both dimensions. When considering only one dimension, the number of weights decreases linearly with d rather than quadratically. We have clarified this statement in the revised paper.
> >
> > **weight decay has an important effect on exploration and used successfully in literature [1,2]. Do you use weight decay in your experiments? Are they included in the gradients when you scale the them for Figure:3?**
> >
> > Convnets typically use some form of weight regularization for image classification tasks. However, we did not employ any additional weight regularization besides what was used for dense training. The motivation was to keep hyperparameters and the training process equivalent to dense, which facilitates adoption. While interesting to explore with sparsity, how weight decay should be applied may vary across deep learning workloads.
> >
> > **I am not sure the illustration on Figure-2 is helpful giving an example where extra dimensions can be helpful. Since even with the vertical dimension, the optimization would stay at the bad local minima for the given example. Maybe "mexican-hat" example would be a better fit.**
> >
> > We removed this example as part of a wider effort to prioritize on the novelty of the contributions in this work, and instead refer to the relevant literature on benefits of overparameterization for escaping local minima.

---

> > > ### Author Response · Authors · 2021-11-23
> > > **Paper4028 Author Responses to Questions from Reviewer t5MH (Part 3/3)**
> > >
> > > **However, authors use half of the paper focusing on explaining why more connections are needed for better results, which I find not very novel and a bit confusing. Authors say that pruned connections are used in training for exploration and search very vaguely; which doesn't provide any new insights on the existing understanding of the training of sparse neural networks.**
> > >
> > > As the novelty of our work in comparison to others has been questioned by several of the reviewers, we made significant changes in our paper to better define these. We also removed most of the early results involving search spaces so we can place a greater focus on the recommendations and empirical results. We hope this addresses the reviewers concern regarding where the emphasis of the paper resides.
> > >
> > > We have addressed most of the minor concerns brought by the reviewer in the revised version of the paper.

---

### Official Review · Reviewer_LwfS · 2021-11-03

**Correctness:** 3
**Technical Novelty And Significance:** 2
**Empirical Novelty And Significance:** 3
**Recommendation:** 5
**Confidence:** 3

**Details Of Ethics Concerns:**

No access to code, lack of reproducibility

**Main Review:**

Strengths:
They provide a very intuitive explanation of the problem with good drawings and figures. This gave me a new intuition about how neural networks learn and optimize
There's a lot of comparisons with current regular neural networks.
The project details are well documented in the appendix
The paper is well written

Weaknesses:
I can't find any comparisons with other, similar, sparsity methods. I would like to see how this holds up against state-of-the-art in sparsity.
I would like to see confidence intervals on the different model performances (if you have the resources of course).
The code is not available (as far as I can see).

**Summary Of The Paper:**

The domain is the optimization of deep and large neural networks. The author argues and presents information that supports (figure 1), a hypothesis that a large portion of neurons is pushed towards dormancy while optimizing a neural network and only a subset of the neurons are actively used. The authors, moreover, argue that these dormant weights are important during optimization as they can propose new solutions during training (figure 2). However, these dormant weights can also be an issue as they introduce gradient noise. The authors propose a new scheme to balance the role of the dormant weights in order to maximize new solutions while minimizing gradient noise. The authors do this by either setting them to zero at a certain iteration or regularizing their sizes. They only optimize a set amount of the weights at once (rewind). The authors show results on multiple benchmark datasets highlighting their performance and how new sparsity-oriented hardware works well with their algorithms. The dormitory vs active weights are picked by a threshold of their magnitude.

**Summary Of The Review:**

I think this is interesting research and highlights a general and contemporary challenge when training deep neural networks. The problem is well motivated and the implementation well described. The experiments are extensive. however, as this is not compared against any other model in the field I cannot assess whether this approach is truly useful. I would be able to change my mind if the authors provide such results, or argue why they don't provide it. Also, please make your code available in an anonymous repository, and publicly available at the end of the review session. If you have the compute, please provide confidence intervals on your results. Also, please elaborate on what it means for weights to "decorrelate". E.g. why is the threshold important.

---

> ### Author Response · Authors · 2021-11-23
> **Paper4028 Author Responses to Questions from Reviewer LwfS (Part 1/1)**
>
> We would like to thank the reviewer for their feedback. We address the reviewer's question below.
>
> **I would like to see how this holds up against state-of-the-art in sparsity.**
>
> We appreciate the reviewer’s desire to compare against other state-of-the-art sparse methods. We do provide some comparisons in Figure 4 against RigL, SET, and Lottery Tickets. These methods were chosen specifically because they use some elements in our recommendations, but omit others that are critical for training sparse neural models. We avoided comparisons with recent advances in sparsity literature [1,2], as they are subsets of the methodology proposed in the paper, and therefore should perform similarly. For example, [1] is shown in FIgure 3 under the curve labeled as “Regularize”. On the other hand, [2] uses 4:8 patterns which are more flexible than 2:4 patterns, and thus should be strictly better, but also are not accelerable on current hardware.
>
> **I would like to see confidence intervals on the different model performances (if you have the resources of course).**
>
> Unfortunately because of the large number of experiments, confidence intervals would require a prohibitive amount of resources. We decided instead to prioritize on validating our method across a broad number of models, tasks, sparsity patterns, and amounts of sparsity, such as to maximize the amount of available information to the research community. At the same time, we believe that the vast number of experiments provide some fashion of confidence intervals for the quality of the method.
>
> **The code is not available (as far as I can see).**
>
> We hope to make the code available near the end of the review process.
>
> **Also, please elaborate on what it means for weights to "decorrelate". E.g. why is the threshold important.**
>
> We use correlations to measure whether weights are learning meaningful representations throughout training. We speculate when weights “decorrelate” they are no longer learning, as they do not have any relation to previous weight values, and can be discarded from training. We choose a correlation threshold below which it is safe to infer correlations are not meaningful. The choice of threshold is important, as a high(low) threshold might over(under)-state when weights stop learning.
>
> [1] A. Zhou, Y. Ma, J. Zhu, J. Liu, Z. Zhang, K. Yuan, W. Sun, , and H. Li. Learning n:m fine-grained structured sparse neural networks from scratch. In ICLR, 2021.
>
> [2] Itay Hubara, Brian Chmiel, Moshe Island, Ron Banner, Seffi Naor, and Daniel Soudry. Accelerated sparse neural training: A provable and efficient method to find n:m transposable masks. arXiv preprint arXiv:2102.08124, 2021.

---

### Official Review · Reviewer_kChj · 2021-11-05

**Correctness:** 3
**Technical Novelty And Significance:** 2
**Empirical Novelty And Significance:** 2
**Recommendation:** 5
**Confidence:** 3

**Main Review:**

Strengths
- This paper is well written and easy to understand
- They make clear the positioning of the paper in relation to prior work and what they aim to demonstrate and compare against.
- The central idea about extra removable weights being extra degrees of freedom is well explained and may be somewhat novel
- They demonstrate significantly better results compared to prior work that does not use frequent "rewiring" with gradient backpropagation on non-contributing weights.
- They demonstrate the effects of the different methods and recommendations with their thorough ablation and exploration of the effects of different hyperparameter values.


Weaknesses
- The novelty of the paper (though admitted by the authors) could be more significant. The method itself does not seem novel and has been explored and used in depth in several other papers.
- These papers are cited, but not compared against. While it is understandable that the main focus of the paper is not on doing significantly better than these methods since they share a similar concept, the paper may be more novel if it can more clearly differentiate their contribution. It is difficult to judge the empirical results in that respect without those comparisons.
- While the related works are mentioned and cited, the central idea could be better compared to the variety of recent work analyzing Deep Neural networks from a loss landscape perspective. It is somewhat difficult to ascertain how novel this central idea is without this comparison. It is quite similar in effect and possibly concept with many other works on analyzing escaping local minima and overparameterized networks.

Questions:
It may be helpful to directly state this in the response especially if I am misunderstanding or underappreciating something. What is the main novel contribution of this paper that differentiates it from other works in this space?
Based on Figure 3, it seems like resetting weights to 0 may have a negative effect on model performance? Doe this trend continue if the length between resets in lengthened even further?

Can you baseline your analysis of the weight movement of active and inactive weights compared to random walks of the model weights? Some of the trends are difficult to differentiate from the numerical properties of the metrics. The trend for active and inactive weights must be monotonically increasing for all the weights due to the definition since at the end of training there is no more movement of the weights. For the correlation, larger weights with a random walk for a fixed step size would naturally require more time to uncorrelate. What is the measure for distance traversed in the second graph in figure 1?

The paper may be improved by exploring the benefits of this sparse training in terms of inference speedup and training speedup.



Minor comments:
"While LOTTERY (Frankle & Carbin, 2019) determines at initialization which weights participate in sparse models"
That cited paper determined based on a full training run based on weight magnitude and then reset the initialization.

Table 1 and 2 would be easier to understand if all the results were consistently error or accuracy or at least if + or - delta was consistently better or worse


**Summary Of The Paper:**

This paper seeks to explore and understand the improved performance of Neural Networks trained with additional parameters even though a majority of the weights can be pruned during inference. They analyze the magnitude, correlation, and movement of weights during training and propose a hypothesis that the reason is that the weights which can be pruned during inference provide degrees of freedom in the loss landscape which allows better optimization of the non-prunable weights. Based on this, they explore and recommend a methodology for sparse training as a method to efficiently approximate a higher dimensional weight space that maintains these good optimization properties. They ablate the effects of the different recommendations and explore the hyperparameter space to have a better understanding of the dynamics of the sparse training.

**Summary Of The Review:**

This paper is well written, and clearly demonstrates the advantages of sparse training with rewiring, gradient updates for non-participating weights, and an induced exploitation step. The paper also includes very thorough and useful empirical results on sparse training and through ablation experiments, but lack comparisons to similar methods. Unfortunately, I currently recommend weak rejection since I don't believe that there are quite enough significant novel contributions. While the experiments and ablations shed some further light into how to effectively use these methods and provides some better understanding, this space and the concepts have been thoroughly explored in previous works.


===============================================================
Post-rebuttal
I thank the authors for their clarifications in the comments and changes in the paper. I believe much of it is clearer now. Due to this, I will be raising my recommendation to marginally below the acceptance threshold. Still believe that as other reviewers have mentioned, the paper would be significantly stronger if they compared their recommendations more directly with other recent work in the space since they claim to be providing extensive hyperparameter recommendations and comprehensive recommendations for how to train sparse models.


There are also minor changes which would make the figures much clearer. Figure 3 should include their method with the combination of  Fix, Reset, and Regularization. The trend with rewiring and resetting steps in Figure 2 would also be clearer if the graphs were slightly extended to demonstrate at what frequency resetting and rewiring is required.

---

> ### Author Response · Authors · 2021-11-23
> **Paper4028 Author Responses to Questions from Reviewer kChj (Part 1/2)**
>
> We thank the reviewer for their detailed and insightful feedback. The reviewer's main concerns seem to surround the novelty of the paper, both regarding the understanding of the importance of pruned weights for training described in Section 2 and the recommendations on how to train sparse models described in Section 3. We have adjusted various sections in the paper to better differentiate our contributions from those of previous works, and summarize these contributions below.
>
> **Understanding the importance of pruned out weights for training.**
>
> While we agree with the reviewer that overparameterization has already been thoroughly studied in literature, we believe our paper introduces several new insights to this topic:
>
> 1. We introduce a new metric (temporal correlation of weights) to measure the interval for which weights retain/learn useful information during training.
>
> 2. We find that weights pruned out at inference do not learn long-term representations, and as such do not need to be stored throughout all of training, but they do learn short-term representations that are needed to explore the parameter space.
>
> 3. We show empirically that these weights can be discarded beyond intervals when their correlations vanish.
>
> The understanding above is fundamental for developing the recommendations we make for sparse training, as well as consolidating and explaining many of the advances in sparsity literature.
>
> **Recommendations on how to train sparse models.**
>
> As pointed out by the reviewer, several previous works have studied one or a combination of the methods outlined in this paper. However, few of these works have been successful in training sparse models from scratch, and even fewer provide an explanation behind their choice of methods and hyperparameters, making their adoption challenging. We believe that our contributions stand out from previous works on several fronts:
>
> 1. The recommendations outlined in this paper are amongst the first to enable training sparse models from scratch (with no dense training) that achieve competitive accuracy to dense counterparts, and are accelerable on existing hardware for both training and inference. While recent works [1,2] have proposed similar methods, neither are accelerable on hardware for training. [1] uses 1D N:M patterns which cannot accelerate the backward pass in training and thus are only useful for inference. [2] uses 4:8 patterns, which does not map to the format used by NVIDIA Sparse Tensor Cores.
>
> 2. We propose new mechanisms to reduce the noise from gradient accumulations beyond what was introduced in [1]. One of them involves resetting non-participating weights to zero every so often. We find that discarding weights at intervals when their correlations vanish (beyond which we conjecture weights are no longer learning) improves accuracy considerably. We also find the optimal resetting interval of 1 epoch to coincide with the regularization factor suggested by [1].
>
> 3. We provide extensive reasoning on the benefits of the methods employed along with guidelines on choices for hyperparameters based on empirical ablations and theoretical deductions. We believe these consolidate the many methods that have been circulating the sparsity literature but have not been widely adopted due to poor hyperparameter choices. For example, several works choose to rewire weights at epoch boundaries, which creates a new direction of optimization in the parameter space only once every epoch, whereas we find that new directions must be formed several orders of magnitudes more frequently. Other studies discard the contents of pruned weights at every rewiring, which we show are critical for learning short-term representations needed to explore the parameter space, which often last for roughly an epoch of training. Lastly, we verify our methods and hyperparameter ablations across dozens of networks and tasks to validate the effectiveness of the guidelines proposed. W e believe the experimental results could be quite useful and impactful to the community, as many of those workloads have not been considered before for sparsity.
>
> While our methodology can train sparse models without losing accuracy with moderate amounts of sparsity (as verified on dozens of networks and tasks) there is visible degradation at higher degrees of sparsity which needs to be further explored. As a result, we would like to emphasize that the goal of this paper is not per se to propose a final solution for training sparse models. Instead, our goal is to provide intuition and reasoning behind choices that work for sparse training, in the hope they will point future research in the right direction.
>
> [1] Learning n:m fine-grained structured sparse neural networks from scratch. In ICLR, 2021.
>
> [2]  Accelerated sparse neural training: A provable and efficient method to find n:m transposable masks. arXiv preprint arXiv:2102.08124, 2021.

---

> > ### Author Response · Authors · 2021-11-23
> > **Paper4028 Author Responses to Questions from Reviewer kChj (Part 2/2)**
> >
> > **The novelty of the paper (though admitted by the authors) could be more significant. The method itself does not seem novel and has been explored and used in depth in several other papers.**
> >
> > We believe our work brings novelty in comparison to other papers on various fronts:
> > 1. We generalize to a wider class of methods for sparse training. For example, there are alternatives to regularizing pruned weights, such as resetting them to zero after correlations vanish or stop rewiring, which are novel and differ from previous works.
> > 2. We are the first to employ sparsity patterns (namely, 2:4 2D) that can accelerate training of sparse models on existing hardware.
> > 3. We provide explanations and intuitions to steps that are necessary for sparse training, giving others the means to devise their own methods.
> > 4. We provide extensive ablations to provide recommendations for hyperparameters often used in sparse training methods.
> > 5. We validate sparse training on a wide array of models and tasks that have not been previously investigated in sparsity literature.
> >
> > **While it is understandable that the main focus of the paper is not on doing significantly better than these methods since they share a similar concept, the paper may be more novel if it can more clearly differentiate their contribution.**
> >
> > While our recommendations are compared to literature in Section 3, we have added a paragraph to Section 6 that summarizes the differences from related works. The literature review should suffice to compare to previous works that are substantially different from ours.
> >
> > **While the related works are mentioned and cited, the central idea could be better compared to the variety of recent work analyzing Deep Neural networks from a loss landscape perspective**
> >
> > We agree with the reviewer that overparameterization helping training has been circulating in literature for some time now. Therefore, we reduced its discussion in Section 2 (and cite relevant literature instead) to give more attention to the novel contributions of this work.
> >
> > **Based on Figure 3, it seems like resetting weights to 0 may have a negative effect on model performance? Does this trend continue if the length between resets in lengthened even further?**
> >
> > Accuracy improves with increasing length between resets up to a certain point, beyond which accuracy worsens considerably (not shown in Figure 2) due noise from gradient accumulations. We find resetting weights is a critical step to maintain task errors small for sparse training (see Figure 3).
> >
> > **Can you baseline your analysis of the weight movement of active and inactive weights compared to random walks of the model weights? The trend for active and inactive weights must be monotonically increasing for all the weights due to the definition since at the end of training there is no more movement of the weights. For the correlation, larger weights with a random walk for a fixed step size would naturally require more time to uncorrelate.**
> >
> > Is the reviewer referring to the fact that weights that decrease over time will induce some bias i the correlation computation? We believe correlations should be insensitive to the fact that weights go to zero for a few reasons. Correlations are computed in overlapping time windows, where each window computes the weight magnitude’s difference from the mean and normalizes with variance. This means that absolute weight magnitudes (are weights large or small) will not impact the correlation metric, and only the weight’s relative difference to the mean of the window will. Since these time windows are much smaller than the total training duration, general learning rate trends that diminish weight updates should also be minimal.
> >
> > **What is the measure for distance traversed in the second graph in figure 1?**
> >
> > The distance is measured as the difference between weights accumulated across time steps. We have removed this figure as a wider reorganization effort to emphasize on the contributions brought by this paper.
> >
> > **The paper may be improved by exploring the benefits of this sparse training in terms of inference speedup and training speedup.**
> >
> > While we acknowledge the benefits of exploring performance with sparsity, and have done so in other venues, we believe its scope is too large to fit within this paper.At the same time we made many choices in the sparse training methodology with performance in mind, i.e. we sparsify all math layers in the network, we sparsify both forward and backward pass matrix multiplications, and we eliminate any need for dense training.
> >
> > **Table 1 and 2 would be easier to understand if all the results were consistently error or accuracy or at least if + or - delta was consistently better or worse**
> >
> > We changed the Tables to report positive delta as being consistently better. It does not make sense to report only errors because that way we lose information about the base accuracy. Conversely, reporting only accuracy would make comparisons harder.

---

### Author Response · Authors · 2021-11-23
**Summary of revised submission.**

Dear reviewers and AC,

A revised submission is now available. We have made numerous structural changes to the paper in order to address the concerns shared by the reviewers. Our changes are summarized below:

**1. We revised most of the paper (title, abstract, introduction, conclusion, to name a few) to better reflect our contributions.**
In an effort to prioritize on the novelty of the contributions in this work, we removed most of the content on search spaces. Instead, we briefly explain why more weights are needed for training in the introduction and cite the relevant literature.

**2. We reduced the results and discussions about search spaces in Section 2.**
We restructured the section to focus on new insights we bring to this topic. Namely, that weights pruned out at inference do not learn long-term representations, but they do have short-term interactions that are needed to explore the parameter space. This understanding provides critical intuition and reasoning behind the methods we (and existing sparsity literature) adopt for sparse training.

**3. Given the extra available text space, we moved the methodology back to the main paper as Section 4.**

**4. We added more comparisons in the related works to highlight the novelty in our methods.**

**5. We removed various figures and appendices from the revision, we also fixed some of the other figures.**

We would also like to emphasize that the goal of our work is two-fold. We want to provide intuition and reasoning behind methods we recommend in the hope it will point future research in the right direction. On the other hand, we want to show that sparse training can be leveraged today with existing sparse hardware, and inspire the sparsity community to further pursue its adoption.

We would appreciate it if reviewers could please take a look at our revised submission. We thank everyone again for their efforts!

Sincerely,

Paper4028 Authors

---

### Decision · Program_Chairs · 2022-01-20

**Decision:**

Reject

**Comment:**

The reviewers were generally split on this paper. On the one hand, reviewers generally appreciated the clear presentation, discussion, and explanations, and the experiments. On the other hand, most reviewers commented on the lack of comparative evaluation to other works, including works that are related conceptually. While the authors have a potentially reasonable argument for omitting such comparisons, in the balance I do not believe that the reviewers were actually convinced by this. Particularly when the novelty of the contribution is not crystal clear, such comparisons are important, so I am inclined to not recommend acceptance at this point (though I acknowledge that the paper is clear borderline and could be accepted).